# A Longitudinal Study on Cognitive Training for Cognitively Preserved Adults in Liguria, Italy

**DOI:** 10.3390/healthcare12030393

**Published:** 2024-02-02

**Authors:** Massimo Veneziano, Maria Francesca Piazza, Ernesto Palummeri, Chiara Paganino, Giovanni Battista Andreoli, Daniela Amicizia, Filippo Ansaldi

**Affiliations:** 1Local Health Unit 3 (ASL3), 16125 Genoa, Italy; massimo.veneziano@asl3.liguria.it (M.V.); chiara.paganino@asl3.liguria.it (C.P.); 2Regional Health Agency of Liguria (ALiSa), 16121 Genoa, Italy; ernesto.palummeri@alisa.liguria.it (E.P.); giovannibattista.andreoli@alisa.liguria.it (G.B.A.); or daniela.amicizia@unige.it (D.A.); or filippo.ansaldi@unige.it (F.A.); 3Department of Health Sciences (DiSSal), University of Genoa, 16132 Genoa, Italy

**Keywords:** memory training, cognitive status, older adults, gender, age, education

## Abstract

In this study, we examined the effects of memory training on cognitive function and depressive symptoms in a cohort of 794 healthy adults aged 50 years or older. Participants were divided into an active intervention group and a passive intervention group, with various cognitive measures assessed over a one-year period. Univariate analysis revealed that the active intervention group consistently outperformed the passive group in measures of memory self-perception (Memory Complaint Questionnaire—MACQ), depressive symptoms (Geriatric Depression Scale—GDS-4), verbal memory and recall ability (A3LP), and verbal fluency (VF). Significant differences in MACQ scores were observed between the two groups at all time points, indicating enhanced memory self-perception in the active group. GDS-4 scores consistently favored the active group, suggesting a reduction in depressive symptoms. A3LP scores demonstrated that the active group had better verbal memory and recall abilities. VF scores consistently favored the active group, indicating superior language skills and cognitive flexibility. Linear regression model and mixed linear regression model reinforced these findings, with highly significant interaction effects observed between the active/passive group, gender, age, education, and time. These effects were particularly pronounced for MACQ and A3LP scores, indicating the combined impact of these factors on memory self-perception and verbal memory. This study highlights the positive impact of memory training intervention on cognitive function and depressive symptoms in older adults and underscores the importance of considering gender, age, and education in cognitive interventions. Notably, these benefits persist for up to six months from the end of the program. The results provide valuable insights into cognitive changes in aging populations and suggest that tailored memory training programs can yield significant improvements.

## 1. Introduction

Population aging is an irreversible global trend, resulting in the inevitable consequence of the demographic transition [1].

As the population’s age continues to rise, susceptibility to diseases also increases, especially chronic-degenerative diseases such as cardio- and cerebrovascular pathologies, psychiatric disorders, and dementia [2].

According to the most recent World Health Organization (WHO) report, the percentage of people aged over 60 is constantly increasing in the world; in 2020 the number of people aged ≥60 was 1 billion, and by 2030, it will increase to 1.4 billion, reaching 2.1 billion by 2050 [1].

This increase at an unprecedented rate is occurring and is expected to accelerate in the upcoming decades, both in industrialized countries and in developing countries. These data are particularly alarming considering the strong correlation between advanced age and the decline in cognitive function [2,3,4].

The greater vulnerability, defined as “fragility”, would expose the elderly to a greater risk of cognitive deterioration; this can lead to the development of what is known as cognitive fragility, which is a clinical construct that would link the psychophysical fragility (including vascular issues, metabolic pathologies, or depression) and cognitive decline, so much so that cognitive fragility is considered a predictive factor of the onset of dementia and an increased risk of mortality [5,6,7,8,9,10].

The abovementioned phenomenon leads to an increase in the needs of the elderly, resulting in an inevitable growth of healthcare expenditures for the elderly (treatments, social care burden, and social security costs), causing the so-called longevity shock [11].

Among the chronic degenerative diseases related to aging, dementia represents one of the major causes of disability and dependence among the elderly population, and it is this pathology that more than others, over the years, has seen a constant increase in terms of prevalence and incidence on a global scale. According to the Global Burden of Disease (GBD) report on Alzheimer’s disease and other dementias in 2021, the number of dementia cases increased from 20.2 million in 1990 to 43.8 million in 2016 (47 million in 2015 according to the WHO 2022) [12].

Every year, it is estimated that there are at least 10 million new cases of dementia in the world, equivalent to one new case every three seconds, and the most recent WHO forecast data indicate that worldwide cases of dementia will increase to 78 million by 2030 and reach 139 million by 2050 [3,13].

Scientific evidence strongly suggests that by reducing risk factors and implementing healthy lifestyle behaviors, it is possible to maintain a healthy brain and significantly decrease the risk of developing dementia [14,15,16,17,18,19].

For several years, scientific evidence has indicated that education, cognitive stimulation [20], and sociocultural activities [21] set the stage for the effectiveness of primary prevention strategies, particularly cognitive stimulation programs. It is within this context that the Memory Training Liguria (MTraiL) project [22], initiated in 2011, stands as a real-world example and pilot project, offering practical insights into the implementation of primary cognitive prevention initiatives. The MTraiL project emerges as a beacon of innovation within the broader framework of global preventive measures, making a significant contribution to dementia prevention in the Liguria Region. This initiative, born out of collaboration between the Liguria Region, municipalities, regional local health units (LHUs), and volunteer associations, has successfully disseminated the benefits of primary prevention among the population. Preliminary results have shown that memory training courses not only improved participants’ cognitive abilities but also enhanced socialization, motivation, self-esteem, and overall mental well-being [22].

Aligned with WHO guidelines, the MTraiL project involves an innovative conceptual vision, transforming dementia from a “static” disease to a “dynamic” pathology, a condition that can be predicted and modified. It focuses on primary prevention interventions, addressing incorrect lifestyles (such as a sedentary lifestyle, smoking, excessive alcohol consumption, and an unbalanced diet) or some diseases as risk factors (such as hypertension, diabetes, obesity, depression, and hypercholesterolemia). Primary prevention gains prominence as a crucial measure in the fight against dementia, aiming to reduce the incidence of dementia and postpone debilitating symptoms, thereby alleviating the strain on healthcare and social and economic systems [23,24,25,26,27].

Education, cognitive, and sociocultural activities, coupled with physical and lifestyle measures, are recognized protective factors [21].

Among the primary prevention strategies, cognitive stimulation programs appear as particularly effective, activating neuroplasticity, a real protective factor for the brain.

Neurogenesis, the self-renewing property of the central nervous system (CNS), is activated in the hippocampal region through associative memory training [28].

Cognitive stimulation interventions trigger neuronal growth, foster cognitive resilience, and increase the ability of the CNS to resist age-related cognitive changes. This resilience may decelerate cognitive decline, even when initial dementia symptoms manifest [29,30,31,32,33,34,35].

This study delves into the hypothesis that the effectiveness of this memory training program for cognitively preserved Ligurian adults aims to induce neuroplastic changes, foster cognitive resilience, and potentially contribute to an overall reduction in dementia incidence. Through this research, we aspire to provide critical insights into proactive interventions for preserving cognitive health amid the escalating challenges of population aging and increasing dementia prevalence.

## 2. Materials and Methods

This study was a longitudinal study investigating the impact of different memory training interventions on the cognitive status of healthy adults aged 50 years or older.

The study was approved by Ligurian Regional Laws n. 1046 and n. 981 (date 14 December 2018 and 20 November 2019, respectively) and the Local Ethics Committee (cod. 400/2011).

The institutional activities of A.Li.Sa. include handling regional healthcare administrative data and conducting epidemiological studies, projects, and research studies to support strategic decision making in healthcare [36].

### 2.1. Participants and Procedure

Subjects who accessed local health units (LHUs) to receive information on health services were enrolled. It is important to note that the participants involved in this study voluntarily approached the LHUs to take part. Their participation was informed and voluntary, without any form of coercion or solicitation.

Information was disseminated through brochures, posters, and oral information sessions. All participants provided written informed consent prior to the assessment, indicating their voluntary participation and understanding of the study’s objectives, in accordance with the Declaration of Helsinki.

No financial compensation was provided to the participants. Interested participants underwent a preliminary screening process to assess their eligibility for memory training courses with a neuropsychological evaluation.

Inclusion criteria were age ≥50 years, good cognitive functioning indicated by a score of 24 or higher on the Mini-Mental State Examination (MMSE) to exclude participants with signs of dementia, and a score ≥7 on the Clock-Drawing Test (CDT) [37,38].

Patients were excluded if they were younger than 50 years, had a history of clinically significant depression, reported psychiatric or neurological disorders, showed signs of sensory deprivation, had a score lower than 24 on the MMSE, or had a score lower than 7 on the CDT.

After this preliminary evaluation, we chose pairs of volunteering persons that were matched for age, sex, and level of education; then, the persons from each pair were randomly assigned to either the “active” or “passive” intervention groups with software randomization, ensuring a balanced distribution between the “active intervention group” engaged in dynamic training sessions and the “passive intervention group” undertaking theoretical cognitive training at home. The randomization process ensured that participants had an equal chance of being assigned to either the active or passive intervention group.

This random assignment allowed us to control for potential confounding variables, contributing to the internal validity of the study.

In the “active intervention group”, participants engaged in active training on memory, logic, and planning, and, in the “passive intervention group”, individuals completed a theoretical cognitive training program at home based on a theoretical–practical manual. Collective sessions were held once a week for a total of 9 consecutive weeks. Each session lasted 90 min and was led by a properly trained and authorized instructor. The training aimed to teach different mnemonic strategies that participants could apply in their daily lives to reduce memory-related difficulties.

Cognitive status was monitored over time at four different stages for the active and passive groups: (i) a preintervention phase, corresponding to the beginning of the course; (ii) a postintervention phase corresponding to the end of the course; (iii) a 6-month postintervention phase; and (iv) a 12-month postintervention phase.

#### 2.1.1. Tools

The neuropsychological battery included the Mini-Mental State Examination (MMSE), the Clock-Drawing Test (CDT), the Memory Complaint Questionnaire (MACQ) to assess memory self-perception, the Geriatric Depression Scale (GDS) for documenting depressive symptoms, and A3LP and verbal fluency (VF) to measure short-term memory, working memory, and processing speed, respectively.

#### 2.1.2. Mini-Mental State Examination (MMSE)

The MMSE is a widely used screening tool to assess cognitive impairment and provide a general overview of a person’s cognitive function. It includes a series of questions and tasks that assess various cognitive domains such as orientation, memory, attention, language, and visuospatial skills [39].

Participants were asked to complete tasks like recalling a list of words, performing simple arithmetic operations, naming objects, and following verbal and written instructions. Scores on the MMSE range from 0 to 30, with higher scores indicating better cognitive function. In many clinical settings, a score below a certain threshold (e.g., 24) might suggest cognitive impairment and could warrant further evaluation for conditions like dementia.

In this study, a score of 24 or higher on the MMSE was used as an inclusion criterion. This means that participants needed to have a certain level of cognitive function to be eligible for the study, excluding those with significant signs of dementia.

#### 2.1.3. Clock-Drawing Test (CDT)

The Clock-Drawing Test is a simple but informative cognitive screening tool that assesses several cognitive functions, including visuospatial abilities, executive functions, and memory [37,38].

Participants were asked to draw a clock face showing a specific time (e.g., “10 past 11”). The test evaluates the accuracy of the drawing, the spatial arrangement of the numbers, and the correct placement of clock hands. Impairments in these tasks could be indicative of cognitive deficits, particularly in relation to certain types of dementia.

In this study, a score of 7 or higher on the Clock-Drawing Test was used as an inclusion criterion. A higher score suggested better performance in this test and indicated a level of cognitive function that is compatible with the study’s aims.

#### 2.1.4. Memory Complaint Questionnaire (MACQ)

The Memory Complaint Questionnaire is used to assess memory self-perception in individuals. It aims to capture how individuals perceive their own memory abilities, which may not always align with objective measures of cognitive performance [40].

The MACQ used in this study was a 7-item scale considering various aspects of subjective memory perception. It included questions about general and specific memory situations. Each of the first five elements was related to specific situations often reported as challenging for those with declining memory. The sixth element measured the overall decline in memory self-perception. The scoring system allowed for the categorization of memory self-perception as excellent, good, or indicating significant memory complaints.

Excellent memory self-perception was indicated with a score < 15; good memory self-perception was categorized as a score between 16 and 25; and finally, a score of 25 and above on the MACQ indicated significant memory complaints that might require further evaluation.

#### 2.1.5. Geriatric Depression Scale (GDS)

The Geriatric Depression Scale is a widely used self-report questionnaire designed to assess depressive symptoms in older adults [41,42].

It helps identify the presence and severity of depressive symptoms, which can be common in older populations. The GDS used in this study was a 4-point scale that explored different aspects of depression. The total score ranged from 0 (indicating no depression) to 4 (indicating significant depression). This tool assists in understanding the participants’ mental well-being and its potential impact on cognitive function.

#### 2.1.6. Test Ability to Recall Words or Recall Test (A3LP)

A3LP is a cognitive assessment tool that measures verbal memory and recall ability. In this study, participants were asked to write or remember as many words as possible. This test evaluates an individual’s ability to recall a list of words or items after a certain period of time. It provides insight into short-term memory and verbal memory functioning [43].

#### 2.1.7. Verbal Fluency (VF)

Verbal fluency is a test commonly used to assess verbal ability and executive function. In the context of this study, participants were asked to generate words starting with specific letters within a limited time of 60 s per letter. This task requires cognitive flexibility, as participants need to quickly switch between generating words based on different initial letters. Verbal fluency provides information about a person’s language skills, cognitive flexibility, and executive functioning [41,44].

### 2.2. Statistical Analysis

The results for the active and passive groups were descriptively summarized through tabulation and graphical representation of means with standard deviations (SDs), medians with interquartile range, 25th and 75th percentiles for continuous variables, and frequency distributions for categorical variables. The active and passive groups were compared in terms of cognitive and functional outcomes (GDS, MACQ, A3LP, and VF) at the baseline, at the end of the course, and at 6 and 12 months of follow-up.

Nonparametric continuous variables, namely years of education, MMSE, CDT, GDS, MACQ, A3LP, and VF, were analyzed using the Mann–Whitney test for independent samples. Finally, gender as a categorical variable was analyzed using the Pearson chi-square test.

Furthermore, differences in cognitive and functional outcomes (GDS, MACQ, A3LP, and VF) before and after training, and at 6 and 12 months of follow-up were evaluated in the active and passive groups, separately, using the Wilcoxon signed-rank test for dependent variables, and a linear regression model was generated to identify the factors associated with cognitive decline and depressive symptoms.

The linear regression model analysis of cognitive measures over time was performed to examine the influence of various associated factors on cognitive performance. Specifically, the contrasts used for the linear regression models included both group (active/passive) and gender. The analysis included measurements at different time points (T0, T1, T2, and T3) and considered the impact of different groups such as the active/passive group, gender, age, and education. A *p*-value less than 0.05 was considered statistically significant. Linear mixed models for repeated measurements (with a significance threshold set at 5%) were used to analyze each outcome measure. Pertaining to this, both group and time were incorporated as fixed effects, while patients were considered as a random effect, and the outcome measure was treated as the dependent variable. Specifically, the contrasts used for the linear mixed models included both group (active/passive) and time.

The assumptions of linear regression models and linear mixed models, crucial for ensuring the reliability of our analyses, were rigorously tested on model residuals, confirming the appropriateness of our statistical approach.

In our study, we addressed various types of variables to conduct a comprehensive analysis. In particular, gender and group (active/passive) were treated as categorical variables. Moving on to continuous variables, we considered several measurements, namely age, years of education, MMSE, CDT, MACQ, GDS-4, A3LP, and VF. These variables were treated as continuous, allowing us to explore relationships and trends in a more detailed manner. Data were analyzed using JMP version 13.0.0 software (SAS Institute, Cary, NC, USA).

## 3. Results

### 3.1. Analysis Results

A total of 794 healthy adults aged 50 years or older participated in this study and were enrolled at the health local unit.

Specifically, 397 recruited subjects were assigned to the active intervention group and participated in the memory training course, with a maximum of 15–20 participants per group.

The other 397 subjects were assigned to the passive intervention group and completed the course only with a theoretical–practical manual of memory training.

Table 1 presents the main characteristics of the participants at the baseline, comparing the active intervention group and the passive intervention group. The two groups did not significantly differ in terms of age, years of education, and cognitive status.

Both the active and passive intervention groups had an equal distribution of males and females, each comprising 50.0% of the participants (n = 108 for each group).

The median age in the overall sample was 72 years, with an interquartile range (IQR) of 68 to 76 years. Both the active and passive intervention groups had a median age of 72 years, with slightly different IQRs (68 to 77 years for the active intervention group and 68 to 76 years for the passive intervention group). The *p*-value (0.6502) indicated that there was no significant difference in age between the two intervention groups.

The median years of education in the overall sample were 8 years, with an IQR of 5 to 13 years. Both the active and passive intervention groups had a median of 8 years of education, with the same IQR of 5 to 13 years. The *p*-value (0.8413) suggested that there was no significant difference in education levels between the two intervention groups.

The overall median MMSE score was 29, with an IQR of 27 to 30. The active and passive intervention groups had the same median MMSE score of 29, with slightly different IQRs (28 to 30 for the active intervention group and 27 to 30 for the passive intervention group). The *p*-value (0.8292) indicated that there was no significant difference in cognitive functioning (as measured by the MMSE) between the two intervention groups.

The overall median CDT score was 9, with an IQR of 8 to 10. The active and passive intervention groups had the same median CDT score of 9, with the same IQR of 8 to 10. The *p*-value (0.6594) suggested that there was no significant difference in cognitive functioning (as measured by the CDT) between the two intervention groups.

### 3.2. Linear Regression Model

Table 2 and Table 3 present the results of the linear regression model of cognitive measures over time and group comparisons. The tables include *p*-values representing the statistical significance of the observed differences. The classical assumption test results of the regression model are reported in the Appendix A).

#### 3.2.1. MACQ (Memory Complaint Questionnaire)

The *p*-values concerning the comparison between the active and passive intervention groups for MACQ scores were consistently highly significant (all less than 0.0001) across all time points (T0, T1, T2, and T3). This revealed notable differences in memory complaints between the active and passive intervention groups throughout the duration of the study.

In relation to the comparison based on “gender”, the *p*-values were below 0.05 at T0, T2, and T3, but they did not reach significance at T1. This implied the presence of gender-based disparities in memory complaints at specific time points.

Contrary to this, the “age” comparison yielded nonsignificant *p*-values across the board, indicating that age did not exert a statistically significant impact on MACQ scores at any of the time points. Similarly, the “education” comparison also displayed *p*-values exceeding 0.05 at very time points, underscoring the absence of significant differences related to education in terms of memory complaints.

#### 3.2.2. GDS-4 (Geriatric Depression Scale)

The *p*-values for the comparison between the active and passive intervention groups for the GDS-4 scores were highly significant (less than 0.0001) at all time points. The “gender” comparison showed nonsignificant *p*-values at all time points, indicating no significant gender-related differences in depressive symptoms. The “age” comparison showed nonsignificant *p*-values at all time points, indicating no significant age-related differences in depressive symptoms. The “education” comparison revealed that there were some education-related differences in depressive symptoms at T0, T1, T2, and T3, with *p*-values less than 0.05.

#### 3.2.3. A3LP (Ability to Recall Words)

The *p*-values for the comparison between the active and passive intervention groups for A3LP scores were highly significant (less than 0.0001) at all time points (T0, T1, T2, and T3). This indicated significant differences in A3LP scores between the two intervention groups throughout the study. The “gender” comparison showed nonsignificant *p*-values at all time points, indicating no significant gender-related differences in A3LP scores. The “age” comparison showed significant differences at T0, suggesting that there might be some age-related differences in A3LP scores at the baseline. The “education” comparison was highly significant (less than 0.0001) at all time points, indicating significant education-related differences in A3LP scores.

#### 3.2.4. VF (Verbal Fluency)

The *p*-values for the comparison between the active and passive intervention groups for the VF scores were 0.0194 at T0 and less than 0.0001 at T1 and T2. This suggested significant differences in VF scores between the two intervention groups at all time points except T3 vs. T0. The “gender” comparison showed nonsignificant *p*-values at all time points, indicating no significant gender-related differences in VF scores. The “age” comparison was highly significant (less than 0.0001) at all time points, indicating significant age-related differences in VF scores. Education level significantly influenced cognitive performance at all time points (*p* < 0.0001 for all comparisons).

### 3.3. Multivariate Analysis of Variance (MANOVA)

Table 4 presents the main significative results of multivariate analysis of variance (MANOVA) for various cognitive measures over time, with different factors analyzed.

The results in Table 4 were obtained from the multivariate analysis of variance (MANOVA) and represented the statistical significance of various interaction effects between different factors (e.g., intervention group, gender, age, education, and time) considering different cognitive measures (MACQ, GDS-4, A3LP, and VF).

#### 3.3.1. MACQ (Memory Complaint Questionnaire)

The *p*-value of <0.0001 indicated a highly significant interaction effect between the “Active/Passive Group” and “Time” on MACQ scores. The associated effect size (F-value) of 197.4111 was substantial, underscoring significant differences between the groups at distinct time points. Regarding gender, the *p*-value of 0.0022 indicated a significant interaction effect with “Time”, and the moderate effect size (F-value = 4.9087) suggested variability in gender-related memory complaints over time. The “Active/Passive Group” * “Gender” * “Time” interaction (*p* = 0.0004, F-value = 6.1096) indicated the combined impact of these factors on memory complaints, warranting further investigation.

#### 3.3.2. GDS-4 (Geriatric Depression Scale)

The *p*-value of <0.0001 reflected a highly significant “Active/Passive Group” * “Time” interaction on GDS-4 scores. However, the effect size (F-value = 1.1502) indicated less pronounced differences between the groups at different time points compared to other comparisons.

#### 3.3.3. A3LP (Ability to Recall Words)

The *p*-value for the “Active/Passive Group * Time” interaction was <0.0001, signifying a highly significant connection between these factors and A3LP scores. With an effect size (F-value) of 92.7409, the substantial difference in scores between the groups at various time points became evident. Moreover, when considering the “Active/Passive Group * Gender * Time” interaction, the *p*-value of 0.0050 indicated a noteworthy three-way interplay among these variables. The effect size (F-value) of 4.3082, characterized as moderate, highlighted the significant collective impact on A3LP scores, which requires further exploration. Analyzing the “Age * Gender * Time” interaction, a *p*-value of 0.0019 unveiled a significant association between age, gender, and time with respect to A3LP scores. The corresponding moderate effect size (F-value = 5.0322) denoted variable influences across different time points. Similarly, the “Gender * Education * Time” interaction yielded a highly significant *p*-value of 0.0008, underlining the joint impact of these factors on A3LP scores. The effect size (F-value) of 5.6587, considered moderate, emphasized the substantial nature of their combined influence. Expanding the analysis, we found that the “Active/Passive Group * Gender * Education * Time” interaction held a *p*-value of 0.0354, signifying significant four-way interplay. The associated moderate effect size (F-value = 2.8749) indicated the need for a deeper investigation into the collective impact. Lastly, the “Active/Passive Group * Gender * Age * Education * Time” interaction featured a *p*-value of 0.0087, unveiling a significant five-way association. The corresponding moderate effect size (F-value = 3.9046) suggested noteworthy, combined influences on A3LP scores.

#### 3.3.4. VF (Verbal Fluency)

Analyzing the “Active/Passive Group * Time” interaction for VF scores, the *p*-value was <0.0001, indicating a highly significant interplay between these factors. The associated effect size (F-value) of 46.7436 suggested substantial score differences between the groups across different time points. The analysis of the “Age * Time” interaction revealed a *p*-value of 0.0006, signifying a meaningful relationship between age and time concerning VF scores. The corresponding moderate effect size (F-value = 5.8422) highlighted age-related variations over different time points. Investigating the “Active/Passive Group * Gender * Age * Time” interaction, the *p*-value was 0.0367, indicating a noteworthy four-way association. The moderate effect size (F-value = 2.8477) accentuated the need for further examination of their collective impact.

### 3.4. Mixed Linear Model

The results of the mixed linear model indicate significant time and group effects for all the variables tested (MACQ, GDS-4, A3LP, and VF), as evidenced by the F-values and the extremely low *p*-values (<0.0001) in each case (Table 5).

### 3.5. Post Hoc Analyses

Table 6 and Figure 1 present the results for the various outcome measures in this study, comparing the active and the passive intervention groups. In particular, the table shows the median values and IQRs for each outcome measure at different time points (T0 = baseline, T1 = after 2 months of rehabilitation, T2 = after 6 months of follow-up, and T3 = after 12 months of follow-up). Additionally, it includes the *p*-values obtained from Mann–Whitney U tests, which were used for comparing the two independent groups (active intervention vs. passive intervention) for nonparametric data.

#### 3.5.1. MACQ (Memory Complaint Questionnaire)

The comparison of MACQ scores between the active and passive intervention groups consistently showed significant differences across various time points. At the baseline (T0), the active group demonstrated lower median MACQ scores (18, IQR: 15–23) than the passive group (23, IQR: 19–26), suggesting a more positive memory self-perception. This trend persisted at subsequent time points (T1, T2, and T3) as well. The active group consistently reported improved memory self-perception with lower MACQ scores, while the passive group maintained higher scores, indicating significant memory complaints.

Comparing the active and passive intervention groups, the MACQ scores showed consistent improvements over time in the active group. Specifically, at T1 (after 2 months of rehabilitation), the active group exhibited a significant reduction in MACQ scores (−5, IQR: −8–2) compared to the baseline (T0), indicating enhanced memory self-perception. This improvement was also evident at T2 (after 6 months of follow-up) with a further reduction in scores (−4, IQR: −7–0), and at T3 (after 12 months of follow-up) with a relatively smaller reduction (−1, IQR: −5–3).

Regarding gender differences, at T0, the difference was significant (*p* = 0.0163) with females having lower scores than males. At the other time points, both males and females had similar median MACQ scores.

The median MACQ scores did not show significant differences between the age groups (≤72 vs. >72) at any time point, and education years did not have a significant impact on MACQ scores at any time point.

#### 3.5.2. GDS-4 (Geriatric Depression Scale)

The GDS scores exhibited notable differences between the active and passive intervention groups. At the baseline (T0), the active group displayed lower median GDS scores (0, IQR: 0–1) than the passive group (1, IQR: 0–2), indicating a relatively lower presence of depressive symptoms. These differences were consistent across subsequent time points (T1, T2, and T3), suggesting that the active group experienced less depressive symptomatology than the passive group throughout the study.

In the active intervention group, the GDS scores consistently remained lower than those in the passive group across time points. Specifically, at T1 (after 2 months of rehabilitation), the active group maintained lower GDS scores (0, IQR: −1–0) than the baseline (T0), indicating a reduction in depressive symptoms. This pattern persisted at T2 (after 6 months of follow-up), with minimal changes (0, IQR: 0–0), and at T3 (after 12 months of follow-up), with a slight increase in scores (1, IQR: 0–1).

No significant gender-based differences in GDS-4 scores were observed at any time point. The age groups did not reveal significant differences in GDS-4 scores and education years did not significantly influence GDS-4 scores at any time point.

#### 3.5.3. A3LP (Ability to Recall Words)

The A3LP scores consistently favored the active intervention group. At the baseline (T0), the active group showed higher median A3LP scores (13, IQR: 10–16) than the passive group (10, IQR: 7–14), indicating better verbal memory and recall ability. This pattern persisted at all follow-up time points (T1, T2, and T3), signifying that the active group consistently outperformed the passive group in verbal memory tasks, suggesting improved memory functioning.

The active intervention group consistently exhibited higher A3LP scores, indicating better verbal memory and recall ability. At T1 (after 2 months of rehabilitation), the active group displayed improved scores (3, IQR: 1–5) compared to the baseline (T0), demonstrating enhanced verbal memory. This trend persisted at T2 (after 6 months of follow-up), with sustained improvement (1, IQR: −3–4), and at T3 (after 12 months of follow-up), with a reduction in scores (−2, IQR: −4–−1).

No notable gender-related differences in A3LP scores were observed at any time point.

Older individuals (>72) generally had lower A3LP scores than younger individuals (≤72), with the differences being statistically significant at T0 (*p* = 0.0005) and T1 (*p* = 0.0227).

Participants with education years >8 had higher A3LP scores than those with education years ≤8. The difference was significant at T0 (*p* = 0.0205), T1 (*p* = 0.0045), T2 (*p* = 0.0073), and T3 (*p* = 0.0009).

#### 3.5.4. VF (Verbal Fluency)

The verbal fluency (VF) scores consistently reflected better performance in the active intervention group. At the baseline (T0), the active group displayed higher median VF scores (25, IQR: 20–31) than the passive group (22, IQR: 16–31), indicating enhanced language skills and cognitive flexibility. These differences persisted at all follow-up time points (T1, T2, and T3), indicating that the active group consistently excelled in generating words based on specific letters within a limited time, reflecting superior cognitive and linguistic abilities.

Furthermore, at T1 (after 2 months of rehabilitation), the active group exhibited higher VF scores (3, IQR: −1–8) than the baseline (T0), indicating improved language skills and cognitive flexibility. This trend continued at T2 (after 6 months of follow-up), with sustained improvement (1, IQR: −3–4), and at T3 (after 12 months of follow-up), with a slight decrease in scores (−3, IQR: −8–1).

No notable gender-related differences in VF scores were observed at any time point. Older individuals (>72) had lower VF scores than younger individuals (≤72), and this difference was significant at all time points (*p* < 0.0001). Participants with education years >8 consistently had higher VF scores than those with education years ≤8. The differences were significant at all time points.

## 4. Discussion

Dementia is characterized by the progressive deterioration of memory, language skills, problem-solving abilities, and other cognitive functions that are severe enough to interfere with daily life. It encompasses a wide range of diseases with over 100 different conditions recognized as potential causes of dementia. As no resolutive therapies are currently available, prevention has become imperative. Indeed, in light of the limited success of conventional pharmacological treatments, prevention measures have gained increasing importance, as endorsed by major scientific organizations and government bodies [14]. There is compelling evidence indicating that adopting a healthy lifestyle and engaging in memory training can help reduce the risk of developing dementia in later life.

A recent systematic review of multidomain lifestyle interventions such as diet, physical activity, and cognitive activities found a significant decrease in the risk of cognitive decline and thus dementia [45].

Among the various primary prevention interventions, cognitive stimulation programs appear to be the most effective, as they most clearly appear to activate the process of neuroplasticity, a crucial factor in brain protection [46].

Our study aimed to investigate the effects of memory training on cognitive and functional outcomes in healthy individuals aged 50 years and above, highlighting the positive impact of prevention intervention on cognitive function.

Similarly, other studies investigated the benefit of cognitive stimulation. A pilot randomized controlled study involving 64 patients without dementia aged 45 to 79 years revealed an improved cognitive status in the intervention group. Specifically, the brain-derived neurotrophic factor (BDNF) was found to be a potential mechanism of the effects of acute exercise on cognitive performance [47].

Identical evidence of improvements in training tasks has been reported among young and elderly adults, showing that systematic training facilitates prospective memory for individuals [48,49].

Furthermore, several meta-analyses of memory training in healthy and older adults reported improved performance immediately after training [50,51,52].

In particular, a meta-analysis investigating the effects of mnemonic training on memory function in healthy older adults examined 21 randomized controlled trials on immediate and long-term effects. The authors found significant immediate and long-term effects. The findings presented relevant practical implications for carrying out further mnemonic training research [53].

In this context, our pilot study evaluated the effectiveness of the implementation of memory training targeted to Ligurian adults and older subjects, carried out by Public Health. This project promoted primary prevention and the importance of having the right lifestyle to keep the brain in good health and reduce the risk of developing dementia.

Indeed, the benefit of WM training in enhancing cognitive–communicative abilities with the help of a structured training program among middle-aged adults was demonstrated. The study outcomes were assessed using standardized neuropsychological tests.

Our study results seem encouraging for the promotion of healthy cognitive well-being among aging adults and provide guidance for the planning and implementation of routine public health activities. The effects of mnemonic training on everyday functioning and the maintenance of training effects over time will provide more detailed data by means of more real-world outcomes and multiple follow-ups.

The novelty of our study in memory is in the broad group of participants involved in the study.

The good performance of MT could not only be observed in the short term, but it could also persist over the long term, and this represents an important starting point for future structured activities carried out by Public Health (Ligurian health units).

In our investigation, no significant differences were found between the two intervention groups in terms of demographic characteristics at the baseline, including gender, age, education, Mini-Mental State Examination (MMSE) scores, and Clock-Drawing Test (CDT) scores.

The results of the linear regression model analysis revealed several significant findings, which provide insights into the impact of the memory training intervention on various cognitive measures.

First, the Memory Complaint Questionnaire (MACQ) scores revealed substantial differences between the intervention groups (*p* <0.0001), indicating a considerable effect of memory training on self-perceived memory abilities. Interestingly, the T3 vs. T0 comparison revealed that the effect of the intervention was sustained at the 12-month follow-up (*p* < 0.0001), suggesting that the benefits of the training endured over time [29].

In terms of demographic factors, gender had a significant influence on the changes in MACQ scores, with females reporting more significant improvements in memory perception than males, as reported in other studies [40].

On the other hand, age and education did not significantly influence the MACQ outcomes, demonstrating the effectiveness of memory training regardless of participants’ age and educational background.

Other authors found different findings; for instance, a low educational level was also associated with higher prevalences of SMC [54].

The second cognitive measure assessed was the Geriatric Depression Scale (GDS), which aimed to evaluate participants’ depressive symptoms. The active/passive comparison showed a highly significant difference, indicating that the memory training intervention led to a reduction in depressive symptoms. Similar to MACQ results, the effect of the intervention on depressive symptoms was maintained at the 12-month follow-up (T3 vs. T0, *p* < 0.0014) [55,56].

Gender and age did not significantly influence the changes in GDS scores, suggesting that memory training had a consistent effect on reducing depressive symptoms across different demographic groups. However, education showed a significant impact, with individuals with higher levels of education exhibiting greater reductions in depressive symptoms.

We also assessed word recall ability, measured by A3LP. The active/passive comparison showed significant improvements in word recall ability due to the memory training intervention (*p* < 0.0001). Moreover, the effect of the intervention was sustained at the 12-month follow-up (T3 vs. T0, *p* < 0.0016), with age and education impacting the outcomes. Older participants and those with less than 8 years of education exhibited less improvement in word recall ability than younger participants and individuals with more than 8 years of education. Gender, however, did not significantly impact the outcomes, indicating that memory training was effective regardless of this demographic factor. The final cognitive measure assessed was verbal fluency (VF), which also showed significant improvements due to the memory training intervention (active/passive, *p* < 0.0001). Similar to the previous cognitive measures, the effect of the intervention on verbal fluency was maintained at the 12-month follow-up (T3 vs. T0, *p* = 0.0144).

Age and education had a significant influence on the changes in verbal fluency scores, with older participants and those with limited education showing less improvement than younger participants and individuals with more than 8 years of education. Gender, however, did not significantly impact the outcomes, indicating that memory training was effective regardless of this demographic factor [57,58].

Overall, the findings from the linear regression and mixed linear models support the efficacy of the memory training intervention in improving cognitive and functional outcomes in healthy older adults. The intervention had a positive impact on self-perceived memory abilities, depressive symptoms, word recall ability, and verbal fluency. Moreover, these effects were sustained over a 12-month period, suggesting the potential for the long-term benefits of training. It is important to note that while the intervention resulted in significant improvements in various cognitive measures, age and educational differences influenced the extent of the improvements. Tailoring future interventions to account for these demographic factors may further enhance the efficacy of memory training programs.

As with other investigations, this study is not exempt from limitations. One major limitation of this study is that it is an observational study and not a randomized controlled trial; therefore, the results of this study can only indicate association, not causation.

The second limitation is the absence of the testing evaluation using functional magnetic resonance imaging, which allows for the detection of any changes at the neuroanatomical level. The third limitation concerns the absence of a clinical outcome enabling the evaluation of the impact of cognitive improvements on daily living skills.

However, our 12-month longitudinal study demonstrates a primary prevention intervention for cognitively preserved people focused on cognitive training, in a socially stimulating environment enriched with light physical activity and the promotion of a healthy diet, that proved effective in improving the performance cognition of the participants and maintaining its effects over time even beyond the end of the program regardless of the initial cognitive state.

Another point of strength was the large sample size, and to the best of our knowledge, this is the first study that evaluated the effectiveness of memory training in a large cohort of individuals.

The current study contributes to the growing body of literature on memory training interventions and their effects on cognitive and functional outcomes in healthy older adults. Further research, including different testing measures and longer follow-up periods, is warranted in order to confirm these findings. Effective memory training interventions could have significant implications for the promotion of cognitive health and well-being in aging populations.

## 5. Conclusions

Our longitudinal study highlights significant improvements in the assessed cognitive performance measures (memory, language, attention, and executive functions) resulting from the proposed cognitive stimulation program. These improvements positively impacted mood and participants’ self-perception of cognitive well-being. Notably, these benefits persisted for up to six months after the end of the program, while some decline was observed at 12 months, except for the enduring improvement in cognitive well-being. The results also reveal that the cognitive response is independent of sociodemographic factors like age and gender but closely linked to education, affirming the protective role of years of schooling in cognitive well-being. The results of the study also suggest that training the memory and all neuropsychological functions is feasible even in old age and that the improvement in cognitive activity, thymic tone, and social well-being are protective factors against physiological brain aging [49,58].

## Figures and Tables

**Figure 1 healthcare-12-00393-f001:**
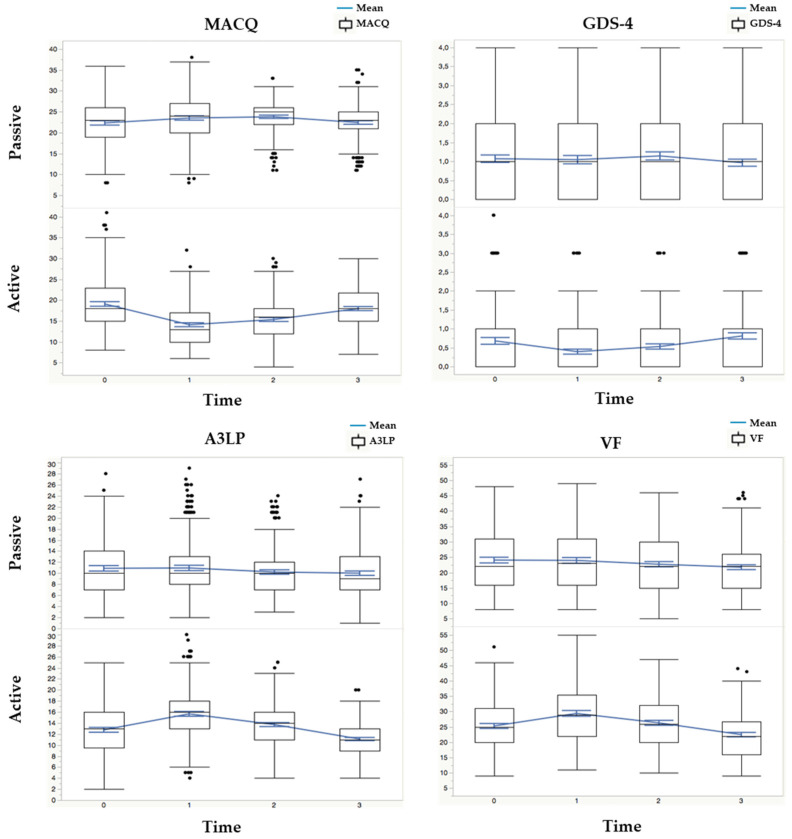
Multipanel figure highlighting the interaction of the time variable (four levels: T0, T1, T2, and T3) with the active/passive group variables for each cognitive and functional measure.

**Table 1 healthcare-12-00393-t001:** Main characteristics of participants at the baseline.

Variables	Overall (n. 794)	Active Intervention Group (n. 397)	Passive Intervention Group (n. 397)	*p*-Value
Gender				
Males (n.; %)	216 (27.2)	108 (50.0)	108 (50.0)	
Females (n.; %)	578 (72.8)	289 (50.0)	289 (50.0)	1.0000 ^‡^
Age (years), median (IQR)	72 (68–76)	72 (68–77)	72 (68–76)	0.6502 ^‡‡^
Education (years), median (IQR)	8 (5–13)	8 (5–13)	8 (5–13)	0.8413 ^‡‡^
MMSE, median (IQR)	29 (27–30)	29 (28–30)	29 (27–30)	0.8292 ^‡‡^
CDT, median (IQR)	9 (8–10)	9 (8–10)	9 (8–10)	0.6594 ^‡‡^

^‡^ Pearson; ^‡‡^ Mann–Whitney.

**Table 2 healthcare-12-00393-t002:** Linear regression model of cognitive measures over time and group comparisons (Part 1).

**MACQ**	**B**	**SE**	** *t* **	***p* T0**	**B**	**SE**	** *t* **	***p* T1**	**B**	**SE**	** *t* **	***p* T2**	**B**	**SE**	** *t* **	***p* T3**
Intercept	22.1672	3.5433	6.2560	<0.0001	22.0753	3.3053	6.6788	<0.0001	20.3483	2.8912	7.0380	<0.0001	21.6145	2.9791	7.2553	<0.0001
Groups (active)	−1.6206	0.1902	−8.5183	<0.0001	−4.6965	0.1775	−26.4641	<0.0001	−4.2233	0.1552	−27.21	<0.0001	−2.2203	0.1600	−13.8760	<0.0001
Gender (females)	−1.0271	0.2138	−4.8031	<0.0001	−0.3561	0.1995	−1.7853	0.0746	−0.4528	0.1745	−2.595	0.0096	−0.5436	0.1798	−3.0233	0.0026
Age	−0.0128	0.0444	−0.2889	0.7728	−0.0439	0.0414	−1.0610	0.2890	−0.0102	0.0362	−0.281	0.7784	−0.0153	0.0373	−0.4088	0.6828
Education	−0.0083	0.0610	−0.1360	0.8918	0.0055	0.0569	0.0968	0.9229	0.01716	0.0497	0.3448	0.7303	−0.0079	0.0513	−0.1542	0.8775
R^2^: 0.1082					R^2^: 0.4725				R^2^: 0.4868				R^2^: 0.2039			
**GDS−4**	**B**	**SE**	** *t* **	***p* T0**	**B**	**SE**	** *t* **	***p* T1**	**B**	**SE**	** *t* **	***p* T2**	**B**	**SE**	** *t* **	***p* T3**
Intercept	0.9735	0.6220	1.5652	0.1179	0.3475	0.6004	0.5787	0.5630	0.7067	0.5974	1.1830	0.2372	0.9125	0.5894	1.5481	0.1220
Groups (active)	−0.1971	0.0334	−5.902	<0.0001	−0.3261	0.0322	−10.1155	<0.0001	−0.3068	0.0321	−9.5670	<0.0001	−0.0784	0.0317	−2.4753	0.0135
Gender (females)	0.0418	0.0375	1.1132	0.2660	0.0515	0.0362	1.4217	0.1555	0.0270	0.0361	0.7501	0.4534	−0.0061	0.0356	−0.1703	0.8648
Age	0.0031	0.0078	0.4015	0.6881	0.0078	0.0075	1.0415	0.2980	0.0045	0.0075	0.5956	0.5516	0.0033	0.0074	0.4402	0.6599
Education	−0.0368	0.0107	−3.4380	0.0006	0.0103	0.0103	−2.2366	0.0256	−0.0217	0.0103	−2.1141	0.0348	−0.0275	0.0101	−2.717	0.0069
R^2^: 0.6380					R^2^: 0.1271				R^2^: 0.1119				R^2^: 0.0222			
**A3LP**	**B**	**SE**	** *t* **	***p* T0**	**B**	**SE**	** *t* **	***p* T1**	**B**	**SE**	** *t* **	***p* T2**	**B**	**SE**	** *t* **	***p* T3**
Intercept	16.6913	3.0103	5.5447	<0.0001	15.3060	2.9696	5.1542	<0.0001	13.747	2.5136	5.4402	<0.0001	9.8348	2.2846	4.3048	<0.0001
Groups (active)	0.9668	0.1616	5.9818	<0.0001	2.3799	0.1594	14.9264	<0.0001	1.7696	0.1350	13.1118	<0.0001	0.5562	0.1227	4.5327	<0.0001
Gender (females)	0.0073	0.1817	0.0404	0.9678	−0.1829	0.1792	−1.0203	0.3079	−0.0815	0.1517	−0.5372	0.5913	−0.0463	0.1379	−0.3359	0.7370
Age	−0.0997	0.0377	−2.6442	0.0084	−0.0596	0.0372	−1.6012	0.1097	−0.0464	0.0315	−1.4724	0.1413	−0.0110	0.0286	−0.3839	0.7012
Education	0.2490	0.0518	4.8071	<0.0001	0.2553	0.0511	4.9955	<0.0001	0.1789	0.0433	4.1370	<0.0001	0.1633	0.0393	4.1508	<0.0001
R^2^: 0.1063					R^2^: 0.2544				R^2^: 0.2058				R^2^: 0.0552			
**VF**	**B**	**SE**	** *t* **	***p* T0**	**B**	**SE**	** *t* **	***p* T1**	**B**	**SE**	** *t* **	***p* T2**	**B**	**SE**	** *t* **	***p* T3**
Intercept	48.9079	5.4429	8.9856	<0.0001	48.2233	5.9950	8.0440	<0.0001	42.5784	5.3635	7.9386	<0.0001	33.1435	4.8788	6.7934	<0.0001
Groups (active)	0.6844	0.2922	2.3419	0.0194	2.7882	0.3219	8.6622	<0.0001	1.8609	0.2880	6.4621	<0.0001	0.4085	0.2620	1.5591	0.1194
Gender (females)	−0.2922	0.3285	0.8895	0.3740	−0.2998	0.3618	−0.8287	0.4075	−0.1109	0.3237	−0.3426	0.7320	0.0193	0.2944	0.0655	0.9478
Age	−0.3910	0.0682	5.7359	<0.0001	−0.3532	0.0751	−4.7034	<0.0001	−0.2981	0.0672	−4.4378	<0.0001	−0.1974	0.0611	−3.3090	0.0013
Education	0.4434	0.0937	4.7348	<0.0001	0.4387	0.1032	4.2531	<0.0001	0.3760	0.0923	4.0746	<0.0001	0.3445	0.0840	4.1006	<0.0001
R^2^: 0.1332					R^2^: 0.1662				R^2^: 0.1282				R^2^: 0.0695			

SE: standard error; T0 = baseline; T1 = after 2 months of rehabilitation; T2 = after 6 months of follow-up; T3 = after 12 months of follow-up.

**Table 3 healthcare-12-00393-t003:** Linear regression model of cognitive measures over time and group comparisons (Part 2).

**Delta MACQ**	**B**	**SE**	** *t* **	***p* T1 vs. T0**	**B**	**SE**	** *t* **	***p* T2 vs. T0**	**B**	**SE**	** *t* **	***p* T3 vs. T0**	**B**	**SE**	** *t* **	***p* T3 vs. T2**
Intercept	−0.0919	2.8809	−0.0319	0.9746	−1.8188	3.1776	−0.5724	0.5672	−0.2654	3.5459	−0.0748	0.9404	1.55341	1.8161	0.8554	0.3926
Groups (active)	−3.0759	0.1547	−19.8856	<0.0001	−2.6027	0.1706	−15.2551	<0.0001	−0.6224	0.1904	−3.2692	0.0011	1.98031	0.0975	20.3088	<0.0001
Gender (females)	0.6709	0.1739	3.8593	0.0001	0.5743	0.1918	2.9946	0.0028	0.4673	0.214	2.1835	0.0293	−0.107	0.1096	−0.9765	0.3291
Age	−0.0311	0.0361	−0.8620	0.3889	0.00263	0.0398	0.0660	0.9474	−0.0055	0.0444	−0.1241	0.9013	−0.0081	0.0227	−0.3578	0.7206
Education	0.0138	0.0496	0.2783	0.7808	0.02545	0.0547	0.4654	0.6417	−0.0082	0.0610	−0.1344	0.8931	−0.0337	0.0313	−1.0768	0.2819
R^2^: 0.3435					R^2^: 0.2349				R^2^: 0.0193				R^2^: 0.3450			
**Delta GDS-4**	**B**	**SE**	** *t* **	***p* T1 vs. T0**	**B**	**SE**	** *t* **	***p* T2 vs. T0**	**B**	**SE**	** *t* **	***p* T3 vs. T0**	**B**	**SE**	** *t* **	***p* T3 vs. T2**
Intercept	−0.626	0.4347	−1.4402	0.1502	−0.2668	0.6568	−0.4063	0.6847	−0.0495	0.6852	−0.0722	0.9424	0.21734	0.4229	0.5139	0.6074
Groups (active)	−0.129	0.0233	−5.5268	<0.0001	−0.1097	0.0353	−3.1116	0.0019	0.1178	0.0368	3.2028	0.0014	0.2275	0.0227	10.0226	<0.0001
Gender (females)	0.0097	0.0262	0.3710	0.7107	−0.0147	0.0396	−0.3719	0.7100	−0.0485	0.0414	−1.1726	0.2413	−0.0338	0.0255	−1.3224	0.1864
Age	0.0047	0.0054	0.8640	0.3879	0.00133	0.0082	0.1614	0.8718	−1.3 × 10^−6^	0.0086	−0.0002	0.9999	−0.0013	0.0053	−0.2510	0.8019
Education	0.0137	0.0075	1.8298	0.0677	0.01506	0.0113	1.3328	0.1830	0.00894	0.0118	0.7582	0.4485	−0.0061	0.0073	−0.8414	0.4004
R^2^: 0.0417					R^2^: 0.0152				R^2^: 0.0154				R^2^: 0.1157			
**Delta A3LP**	**B**	**SE**	** *t* **	***p* T1 vs. T0**	**B**	**SE**	** *t* **	***p* T2 vs. T0**	**B**	**SE**	** *t* **	***p* T3 vs. T0**	**B**	**SE**	** *t* **	***p* T3 vs. T2**
Intercept	−1.3853	2.0166	−0.6870	0.4923	−3.0166	2.1884	−1.3784	0.1685	−6.6665	2.5099	−2.6561	0.0081	−3.6499	1.4313	−2.5500	0.0110
Groups (active)	1.41308	0.1083	13.0508	<0.0001	0.80275	0.1175	6.8318	<0.0001	−0.4256	0.1348	−3.1585	0.0016	−1.2284	0.0769	−15.9841	<0.0001
Gender (females)	−0.1902	0.1217	−1.5627	0.1185	−0.0888	0.1321	−0.6726	0.5014	−0.0644	0.1515	−0.4255	0.6706	0.02439	0.0864	0.2824	0.7777
Age	0.04014	0.0253	1.5892	0.1124	0.05334	0.0274	1.9460	0.0520	0.08668	0.0314	2.7572	0.0060	0.03334	0.0179	1.8595	0.0633
Education	0.00626	0.0347	0.1804	0.8569	−0.0701	0.0377	−1.8607	0.0632	−0.0914	0.0432	−2.1166	0.0346	−0.0213	0.0246	−0.8667	0.3864
R^2^: 0.1829					R^2^: 0.0744				R^2^: 0.042077				R^2^: 0.2494			
**Delta VF**	**B**	**SE**	** *t* **	***p* T1 vs. T0**	**B**	**SE**	** *t* **	***p* T2 vs. T0**	**B**	**SE**	** *t* **	***p* T3 vs. T0**	**B**	SE	** *t* **	***p* T3 vs. T2**
Intercept	−0.6846	3.533	−0.1938	0.8464	−6.3295	3.7153	−1.7036	0.0888	−15.376	4.2359	−3.6301	0.0003	−9.0469	2.5227	−3.5862	0.0004
Groups (active)	2.10382	0.1897	11.0905	<0.0001	1.17655	0.1995	5.8980	<0.0001	−0.3065	0.2274	−1.3475	0.1782	−1.483	0.1355	−10.9488	<0.0001
Gender (females)	−0.0076	0.2132	−0.0358	0.9714	0.18129	0.2242	0.8085	0.4190	0.28946	0.2556	1.1323	0.2579	0.10817	0.1522	0.7105	0.4776
Age	0.03787	0.0443	0.8557	0.3924	0.09291	0.0465	1.9966	0.0462	0.18947	0.0531	3.5710	0.0004	0.09655	0.0316	3.0556	0.0023
Education	−0.0047	0.0608	−0.0775	0.9382	−0.0674	0.0639	−1.0543	0.2921	−0.1106	0.0729	−1.5172	0.1296	−0.0432	0.0434	−0.9948	0.3201
R^2^: 0.1363					R^2^: 0.0554				R^2^: 0.0384				R^2^: 0.1483			

SE: standard error; T0 = baseline; T1 = after 2 months of rehabilitation; T2 = after 6 months of follow-up; T3 = after 12 months of follow-up.

**Table 4 healthcare-12-00393-t004:** Multivariate analysis of variance (MANOVA).

	Value	F Value	NumDF	DenDF	Prob > F
**MACQ**					
Active/Passive Group * Time	0.76411721	197.4111	3	775	<0.0001
Gender * Time	0.0190013	4.9087	3	775	0.0022
Active/Passive Group * Gender * Time	0.02365	6.1096	3	775	0.0004
**GDS-4**					
Active/Passive Group * Time	0.0931533	1.1502	3	775	<0.0001
**A3LP**					
Active/Passive Group * Time	0.3589968	92.7409	3	775	<0.0001
Active/Passive Group * Gender * Time	0.0166771	4.3082	3	775	0.0050
Age * Gender * Time	0.0194795	5.0322	3	775	0.0019
Gender * Education * Time	0.0219046	5.6587	3	775	0.0008
Active/Passive Group * Gender * Education * Time	0.0111288	2.8749	3	775	0.0354
Active/Passive Group * Gender * Age * Education * Time	0.0151146	3.9046	3	775	0.0087
**VF**					
Active/Passive Group * Time	0.1809429	46.7436	3	775	<0.0001
Age * Time	0.0226151	5.8422	3	775	0.0006
Active/Passive Group * Gender * Age * Time	0.0110234	2.8477	3	775	0.0367

* The asterisk denotes an interaction between the variables.

**Table 5 healthcare-12-00393-t005:** Linear mixed model analysis results for time and group effects.

	Time Effect		Group Effect	
	F	*p*-Value	F	*p*-Value
MACQ	57.581	<0.0001	488,275	<0.0001
GDS-4	11.433	<0.0001	72.246	<0.0001
A3LP	222.038	<0.0001	108.156	<0.0001
VF	208.663	<0.0001	23.425	<0.0001

**Table 6 healthcare-12-00393-t006:** Comparison of cognitive and functional measures between participants at baseline and over time expressed as medians and interquartile range (IQR).

	Active Group (n. 397)	Passive Group (n. 397)	*p*-Value	Males (n. 216)	Females (n. 578)	*p*-Value	Age (≤72)	Age (>72)	*p*-Value	Ed. Years (≤8)	Ed. Years (>8)	*p*-Value
**MACQ**												
T0	18 (15–23)	23 (19–26)	<0.0001	22 (18–26)	20 (15–24)	0.0163	21 (16–25)	20 (16–24)	0.1590	21 (16–24)	21 (16–24)	0.1619
T1	13 (10–17)	24 (20–27)	<0.0001	19 (14–24)	18 (12–25)	0.5925	19 (13–25)	18 (12–24)	0.2356	18 (13–24)	19 (13–25)	0.4545
T2	16 (12–18)	25 (22–26)	<0.0001	20 (16–25)	19 (15–25)	0.1959	20 (15–25)	20 (15–25)	0.9375	20 (15–25)	19 (15–25)	0.8202
T3	18 (15–22)	23 (21–25)	<0.0001	22 (18–24)	21 (16–24)	0.0211	21 (16–24)	21 (16–24)	0.9364	21 (16–24)	21 (16–24)	0.5904
**GDS-4**												
T0	0 (0–1)	1 (0–2)	<0.0001	1 (0–1)	1 (0–1)	0.1082	1 (0–1)	1 (0–2)	0.0558	1 (0–2)	0 (0–1)	0.0016
T1	0 (0–1)	1 (0–2)	<0.0001	1 (0–1)	0 (0–1)	0.2938	0 (0–1)	0 (0–1)	0.0776	0 (0–1)	0 (0–1)	0.0944
T2	0 (0–1)	1 (0–2)	<0.0001	1 (0–1)	1 (0–1)	0.9749	1 (0–1)	1 (0–1)	0.2179	1 (0–1)	1 (0–1)	0.1641
T3	1 (0–1)	1 (0–2)	0.0992	1 (0–1)	1 (0–1)	0.2685	1 (0–1)	1 (0–1)	0.3770	1 (0–1)	1 (0–1)	0.0386
**A3LP**												
T0	13 (10–16)	10 (7–14)	<0.0001	12 (9–15)	12 (8–15)	0.8704	12 (9–16)	11 (7–14)	0.0005	11 (8–15)	12 (9–16)	0.0205
T1	16 (13–18)	10 (8–13)	<0.0001	13 (9–17)	13 (9–17)	0.7950	14 (10–17)	12 (8–17)	0.0227	12 (9–16)	15 (10–18)	0.0045
T2	14 (11–16)	10 (7–12)	<0.0001	12 (8–15)	12 (8–15)	0.8732	12 (9–15)	12 (8–15)	0.1372	12 (8–15)	13 (9–15)	0.0073
T3	11 (9–13)	9 (7–13)	<0.0001	10 (8–13)	10 (8–13)	0.8544	11 (8–13)	10 (8–13)	0.0670	10 (8–13)	12 (8–13)	0.0009
**VF**												
T0	25 (20–31)	22 (16–31)	0.0013	24 (18–32)	24 (18–31)	0.9784	26 (21–34)	21 (15–26)	<0.0001	23 (17–29)	26 (21–34)	<0.0001
T1	29 (22–36)	23 (16–31)	<0.0001	27 (19–34)	26 (19–33)	0.1777	29 (22–35)	22 (16–31)	<0.0001	24 (18–32)	29 (22–35)	0.0002
T2	26 (20–32)	22 (15–30)	<0.0001	24 (17–31)	23 (17–30)	0.5958	26 (21–32)	21 (15–28)	<0.0001	23 (16–30)	26 (20–33)	0.0004
T3	22 (16–27)	22 (15–26)	0.6220	22 (15–27)	22 (16–26)	0.3484	23 (18–28)	21 (14–25)	<0.0001	21 (15–25)	23 (18–29)	<0.0001

Ed. Years: education years; T0 = baseline; T1 = after 2 months of rehabilitation; T2 = after 6 months of follow-up; T3 = after 12 months of follow-up.

## Data Availability

The data are not publicly available due to privacy and ethical restrictions.

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
