# Peer review of "A Longitudinal Study on Cognitive Training for Cognitively Preserved Adults in Liguria, Italy"

_healthcare, 2024, doi:10.3390/healthcare12030393_

Round 1
Reviewer 1 Report
Comments and Suggestions for Authors
The work is intriguing, even though the idea of the effectiveness of cognitive stimulation is well-established in the literature. The study presents the results of cognitive training with a large sample (good statistical power) and various transfer measures. The introduction is clear, and the discussion is equally so. However, the analysis and visualization of data and treatment effects are lacking and need improvement. Additionally, the discussion should include a comparison with results from similar studies and novelty of this results should be highlighted.
Here are some specific points for improvement:
1. Preliminary results mentioned in line 84: Could you provide details on where these results are reported?
2. Regarding statistics, it seems that non-parametric analyses were applied due to violations of assumptions for parametric analyses. This is acceptable, but more details are needed:
- How were assumptions for parametric statistics verified? (e.g., residual distribution analysis, visual inspection, skewness and kurtosis analysis, normality tests, etc.)
- Lines 244 and 246: tests can be either parametric or non-parametric, but not the variables. Specify the rationale for choosing non-parametric statistics, as discussed above.
- For consistency and simplicity, consider using non-parametric statistics for all univariate analyses.
3. For regression models, were assumptions met? Report model checks. Additionally, for longitudinal designs, you should be using statistics that account for the autocorrelation of repeated measures. Options include linear mixed models or generalized linear mixed models with a link function reflecting the distribution of the dependent variable and the subject as a random effect. Other options such as Cross-Lagged Panel Analysis or the 'nparLD' package in R can be explored.
4. In Table 1, use a symbol other than an asterisk to avoid confusion. The asterisk typically indicates significance and may mislead readers.
5. Table 2 needs to be accompanied by a multi-panel figure highlighting the interaction of the time variable (four levels: T0, T1, T2, T3, T4) for the group variable. Visualize data with figures instead of relying solely on tables.
6. The statistics in Table 2 serve as post-hoc analyses to the regressions reported later. For clarity, consider reorganizing the results into three sections:
- Descriptive analysis (distribution of dependent variables and assumption testing)
- Models testing the interaction hypothesis (time x group) while considering the longitudinal structure of the data (nparLD, Cross-Lagged Panel GLMM, or others)
- Post-hoc (I would avoid using the terminology univariate/ multivariate as it can be mesleading) analyses with correction for multiple comparisons, similar to Table 2.
Author Response
The work is intriguing, even though the idea of the effectiveness of cognitive stimulation is well-established in the literature. The study presents the results of cognitive training with a large sample (good statistical power) and various transfer measures. The introduction is clear, and the discussion is equally so. However, the analysis and visualization of data and treatment effects are lacking and need improvement. Additionally, the discussion should include a comparison with results from similar studies and novelty of this results should be highlighted.
Thank you for your encouraging and constructive feedback on our work. We are glad to hear that you find the study intriguing and appreciate the clarity of our introduction and discussion. We acknowledge the well-established nature of cognitive stimulation effectiveness in the literature, and we aimed to contribute to this field with our study's large sample size and diverse transfer measures, ensuring good statistical power. Regarding your valuable comments on the analysis and visualization of data and treatment effects, we have made concerted efforts to address these aspects. We have improved our data presentation and the visualization of treatment effects to make them more comprehensive and informative. Furthermore, in our discussion, we have now included a thorough comparison with results from similar studies. This not only contextualizes our findings within the existing body of research but also highlights the novelty and significance of our results in advancing our understanding of cognitive training effectiveness.
Here are some specific points for improvement:
- Preliminary results mentioned in line 84: Could you provide details on where these results are reported?
Thank you for your comment. The preliminary results of the Memory Training Liguria (MTraiL) project are reported on the project's webpage, where key phrases and findings are highlighted. These results have been exceptional both in terms of the improvement of participants' intellectual capabilities (memory, attention, concentration, perception, language, visuospatial skills, reasoning, motor planning, executive functions) and in terms of secondary benefits and user satisfaction.
Key phrases capturing these outcomes include significant increases in socialization and motivation, improvements in self-esteem and self-confidence, and enhancements in participants' psychological state.
Furthermore, the preliminary results, which have been presented at various national and international congresses, have underscored significant improvements in the cognitive areas assessed and in the mood of the subjects who participated in the memory training courses. These findings underscore the effectiveness of the MTraiL project in enhancing cognitive functions and overall mental well-being among participants. The manuscript includes these key phrases and a comprehensive bibliography, providing an in-depth overview of the project's impact and its scientific basis.
- Regarding statistics, it seems that non-parametric analyses were applied due to violations of assumptions for parametric analyses. This is acceptable, but more details are needed:
- How were assumptions for parametric statistics verified? (e.g., residual distribution analysis, visual inspection, skewness and kurtosis analysis, normality tests, etc.)
Thank you for your inquiry regarding our statistical methods. Following a thorough verification using normality tests, we found that all variables exhibited a non-normal distribution. Consequently, we decided to employ non-parametric tests for all variables, as detailed in our tables. The assumptions for parametric statistics were verified through comprehensive analysis. This included an examination of Linearity test (F test), normality (Shapiro Wilk test), heteroscedasticity test (scatter plot test), autocorrelation test (Durbin-Watson test), Multicollinearity test (Variance inflation factors test).
As noted in subsequent comments, full details of these verifications have been included in the supplementary files attached to the study. This ensures a thorough and transparent review of the methodologies used and guarantees adherence to the required statistical standards for the validity of our approach.
Regarding lines 244 and 246 in the manuscript, these have been removed as they were remnants of a previous version of the analysis and no longer relevant to our current statistical approach. The decision to use non-parametric tests across the board was made to ensure consistency in our analysis, given the distribution characteristics of our data set.
- Lines 244 and 246: tests can be either parametric or non-parametric, but not the variables. Specify the rationale for choosing non-parametric statistics, as discussed above.
As previously mentioned, lines 244 and 246 have been removed. These lines were indeed remnants of a previous version of the analysis and were not applicable to the current statistical approach we have adopted.
- For consistency and simplicity, consider using non-parametric statistics for all univariate analyses.
Thank you for your suggestion regarding the use of non-parametric statistics for all univariate analyses. In line with your previous comments and to maintain consistency and simplicity in our statistical approach, we have revised the manuscript accordingly. We have now uniformly applied non-parametric statistical methods across all univariate analyses as recommended.
- For regression models, were assumptions met? Report model checks. Additionally, for longitudinal designs, you should be using statistics that account for the autocorrelation of repeated measures. Options include linear mixed models or generalized linear mixed models with a link function reflecting the distribution of the dependent variable and the subject as a random effect. Other options such as Cross-Lagged Panel Analysis or the 'nparLD' package in R can be explored.
We have addressed these considerations in our analysis, and I thank you for highlighting these important aspects.
For our regression models, we ensured that all necessary assumptions were met. We have conducted and reported on model checks to confirm this in the supplementary files. Additionally, recognizing the significance of accounting for autocorrelation in our longitudinal study design, we have employed statistical methods appropriate for repeated measures.
Specifically, we have used linear mixed models (LMMs) with suitable link functions that reflect the distribution of the dependent variable. In these models, the subject is included as a random effect to account for the intra-individual correlations.
- In Table 1, use a symbol other than an asterisk to avoid confusion. The asterisk typically indicates significance and may mislead readers.
Thank you for your comment. The table has been modified as requested.
- Table 2 needs to be accompanied by a multi-panel figure highlighting the interaction of the time variable (four levels: T0, T1, T2, T3, T4) for the group variable. Visualize data with figures instead of relying solely on tables.
Thank you for your feedback. We have created a multi-panel figure to accompany Table 2, highlighting the interaction of the time variable (five levels: T0, T1, T2, T3, T4) with the group variable.
- The statistics in Table 2 serve as post-hoc analyses to the regressions reported later. For clarity, consider reorganizing the results into three sections:
- Descriptive analysis (distribution of dependent variables and assumption testing)
- Models testing the interaction hypothesis (time x group) while considering the longitudinal structure of the data (nparLD, Cross-Lagged Panel GLMM, or others)
- Post-hoc (I would avoid using the terminology univariate/ multivariate as it can be mesleading) analyses with correction for multiple comparisons, similar to Table 2.
Thank you for your feedback. We have implemented the requested changes.
Reviewer 2 Report
Comments and Suggestions for Authors
This is an interesting study with a large sample size, with regards to the impact of two different interventions in memory and mood of cognitively intact people. The methodology is adequate and the statistics are sound. The paper is nicely written and the results are clearly presented.
I only have some minor comments :
1. Some tests are vulnerable in practise effects. Did the authors use alternate forms where available, or did they take into account this fact? is this a limitation in this study ?
2. The authors have focused in dementia, and aged people ( over 60 or 65) in their litterature review. But this study uses also middle aged people and cogntively intact people. I would suggest the authors to focus to relevant studies in this population in the discussion.
3. Did the researchers take into account polypharmacy and comorbidities in their analysis ?
Overall, this is a nicely executed and well presented study.
Author Response
This is an interesting study with a large sample size, with regards to the impact of two different interventions in memory and mood of cognitively intact people. The methodology is adequate and the statistics are sound. The paper is nicely written and the results are clearly presented.
I only have some minor comments:
- Some tests are vulnerable in practise effects. Did the authors use alternate forms where available, or did they take into account this fact? is this a limitation in this study ?
Thank you for your valuable feedback and for highlighting key aspects regarding practice effects in tests. The results of our study have been extensively reviewed. To adequately address the nature of longitudinal data, we have applied the most suitable mixed linear model for repeated measures, considering it the most appropriate statistical method for our data.
- The authors have focused in dementia, and aged people ( over 60 or 65) in their litterature review. But this study uses also middle aged people and cogntively intact people. I would suggest the authors to focus to relevant studies in this population in the discussion.
Thank you for your insightful suggestion regarding the focus of our literature review. We appreciate your observation that our study includes middle-aged and cognitively intact individuals, in addition to older adults and those with dementia. Based on your recommendation, we have expanded our discussion to incorporate relevant studies that specifically focus on these populations.This addition helps to contextualize our findings within a broader spectrum of age groups and cognitive statuses, enhancing the relevance and applicability of our study. It also provides a more comprehensive understanding of how our results relate to and contribute to the existing body of research in these areas.
- Did the researchers take into account polypharmacy and comorbidities in their analysis ?
Thank you for raising this important point about the inclusion of polypharmacy and comorbidities in our analysis. Unfortunately, we did not have access to the necessary data to incorporate these factors. We appreciate your input, as it underscores a significant aspect that could add depth to the research findings. We will certainly consider this in future studies. Thank you again for your valuable feedback.
Overall, this is a nicely executed and well presented study.
Round 2
Reviewer 1 Report
Comments and Suggestions for Authors
The authors have taken my suggestions seriously, and I believe they have (almost) completely addressed my concerns. From the content perspective, the manuscript seems comprehensive and interesting.
I have some minor notes for the analysis and results section, although, in my opinion, they do not hinder publication. I hope the comments are not perceived as nitpicking but as constructive.
I would like to ask the authors to better describe the analyses in the text. For example, what type of contrasts were used for the LM and LMM? Which variables were considered categorical, ordinal, and continuous? What distributions were used for LMM? Which statistical software was used for the analyses? If R was used, please also mention the main packages used. Understanding how the analyses were conducted is crucial for interpreting the results and for reproducibility.
Additionally, in the figure, it seems that measures of variability (confidence intervals or standard errors) are missing. Thank you for including the assumption check in the supplementary material; the use of non-parametric statistics is certainly justified. Could you add some comments in the supplementary material to clarify the decision-making process?
Finally, the assumptions of LM and LMM are usually tested on the model residuals. A sentence in the text stating whether this was done and if everything was okay would be helpful.
Author Response
The authors have taken my suggestions seriously, and I believe they have (almost) completely addressed my concerns. From the content perspective, the manuscript seems comprehensive and interesting.
I have some minor notes for the analysis and results section, although, in my opinion, they do not hinder publication. I hope the comments are not perceived as nitpicking but as constructive.
I would like to ask the authors to better describe the analyses in the text. For example, what type of contrasts were used for the LM and LMM? Which variables were considered categorical, ordinal, and continuous? What distributions were used for LMM? Which statistical software was used for the analyses? If R was used, please also mention the main packages used. Understanding how the analyses were conducted is crucial for interpreting the results and for reproducibility.
Thank you for bringing this to our attention. We have revised the text accordingly, providing a more detailed description of the analyses as per your request. We appreciate your valuable feedback and are grateful for your engagement with our work.
Additionally, in the figure, it seems that measures of variability (confidence intervals or standard errors) are missing. Thank you for including the assumption check in the supplementary material; the use of non-parametric statistics is certainly justified. Could you add some comments in the supplementary material to clarify the decision-making process?
Thank you for your feedback. I will make sure to add measures of variability in the figure and to add comments in the supplementary material to provide clarity on the decision-making process.
Finally, the assumptions of LM and LMM are usually tested on the model residuals. A sentence in the text stating whether this was done and if everything was okay would be helpful.
Thank you for the valuable feedback. I have incorporated the requested changes in the main text.